# Promising Opportunities for Treating Neurodegenerative Diseases with Mesenchymal Stem Cell-Derived Exosomes

**DOI:** 10.3390/biom10091320

**Published:** 2020-09-15

**Authors:** Reut Guy, Daniel Offen

**Affiliations:** Felsenstein Medical Research Center, Department of Human Molecular Genetics and Biochemistry, Sackler School of Medicine, Tel Aviv University, Tel Aviv 6997801, Israel; rg.reutguy@gmail.com

**Keywords:** mesenchymal stem cell-derived exosomes, neurodegenerative diseases, cell-based therapies

## Abstract

Neurodegenerative disease refers to any pathological condition in which there is a progressive decline in neuronal function resulting from brain atrophy. Despite the immense efforts invested over recent decades in developing treatments for neurodegenerative diseases, effective therapy for these conditions is still an unmet need. One of the promising options for promoting brain recovery and regeneration is mesenchymal stem cell (MSC) transplantation. The therapeutic effect of MSCs is thought to be mediated by their secretome, and specifically, by their exosomes. Research shows that MSC-derived exosomes retain some of the characteristics of their parent MSCs, such as immune system modulation, regulation of neurite outgrowth, promotion of angiogenesis, and the ability to repair damaged tissue. Here, we summarize the functional outcomes observed in animal models of neurodegenerative diseases following MSC-derived exosome treatment. We will examine the proposed mechanisms of action through which MSC-derived exosomes mediate their therapeutic effects and review advanced studies that attempt to enhance the improvement achieved using MSC-derived exosome treatment, with a view towards future clinical use.

## 1. Introduction

Mesenchymal stem cells or mesenchymal stromal cells (MSCs) are self-renewing populations of adult multipotent progenitor cells with the potential to differentiate into several mesodermal cell lineages including bone, cartilage, and adipose tissue [1]. This feature underlies attempts to use MSCs as a therapeutic tool.

The therapeutic potential of MSCs has been tested over the years, in both preclinical and clinical trials for a wide variety of diseases including myocardial infarction, acute renal failure, osteoporosis, type I diabetes mellitus, and pulmonary fibrosis [2,3]. MSC transplantation in neurodegenerative disease models has led to improvement in various parameters, including improved survival, decreased pathology, and rescue of deteriorated cognition [4,5,6]. This is thought to be achieved by the secretion of neurotrophic factors and immunomodulation and within the brain, also by neurogenesis and prevention of misfolded protein aggregation [4].

Despite the positive results obtained with MSCs in therapy, introducing foreign living cells into the human body is always a cause for concern. Exogenously administered MSCs may elicit adverse effects, e.g., immune reactions [7,8,9], embolic phenomena [10], graft versus host disease [11], secondary infection [12], and the risk of malignancy [13,14]. In this context, only a small portion of transplanted MSCs apparently localize to the site of damage and the surrounding area, while most MSCs accumulate in the liver, spleen, and lungs [15]. While MSCs are rapidly cleared from the body following systemic transplantation, their therapeutic benefits typically persist [16]. This has been interpreted to imply that the therapeutic effect is mediated by the MSC-secretome, and specifically by exosomes [17,18].

In this review, we will present advances in MSC-derived exosome-based therapies in models of neurodegenerative diseases such as Alzheimer’s disease, multiple sclerosis (MS), and acute models of stroke. We will examine the clinical outcomes achieved in various neurodegenerative disease models as a result of MSC-derived exosome treatment and describe the proposed mechanisms by which these results are achieved. We will also discuss the limitations of treatment with exosomes and the possibility of improving treatment efficiency in order to transition to clinical trials.

## 2. MSC-Derived Exosomes as a Therapeutic Tool

Extracellular vesicles (EVs) is a general term for a heterogeneous population of 20–1000 nm membranous components that are secreted from both prokaryotic and eukaryotic cells [19,20]. Exosomes comprise a subpopulation of 30–150 nm vesicles containing proteins, mRNA, miRNA, lipids, and DNA that play an important role in intercellular communication via transfer of their content. Exosomes are known to retain the characteristics of the cells from which they are derived [21]. For example, exosomes derived from MSCs, are known for their ability to modulate the immune system, stimulate cell proliferation, promote angiogenesis, prevent apoptosis, and suppress oxidative stress [20].

These traits have been demonstrated in numerous animal studies as having therapeutic potential in a wide range of diseases. Treatment with MSC-derived EVs decreased renal oxidative stress, increased renal cell proliferation, attenuated apoptosis and fibrosis, and normalized renal function in acute kidney injury (AKI) [22]. Administering MSC-derived exosomes improved osteoporosis by promoting the proliferation of osteoblasts via the MAPK pathway [23], while prophylactic treatment with MSC-derived exosomes improved oxidative stress injury and suppressing inflammatory response in traumatic acute lung injury [24]. Furthermore, MSC-derived exosomes are protected against myocardial infarction by promoting autophagy and suppressing apoptosis [25]. Notably, MSCs and MSC-derived exosomes were comparable to one other in reducing inflammation, oxidative stress, and functional deterioration [26,27].

## 3. Clinical Outcomes following MSC-Derived Exosome Treatment in Neurodegenerative Animal Models

Given the success of treatment with exosomes in a variety of diseases, it is not surprising that the possibility of treating neurodegenerative diseases has also been examined. The therapeutic potential of MSC-derived exosomes has been examined in a number of models of neurodegenerative diseases, including Alzheimer’s disease (AD), multiple sclerosis (MS), stroke, neuroinflammation, traumatic brain injury (TBI), spinal cord injury (SCI), and status epilepticus (SE). Exosomes have been shown therapeutic promise in all of these diseases, as reflected by changes in various parameters.

Improvement in functional outcome was observed in stroke, MS, and SCI [28,29,30,31,32,33,34,35,36,37,38,39]. For example, cell death, a common manifestation of neurodegenerative diseases, was reduced by exosome treatment in models of stroke, TBI, perinatal brain injury, and SCI [32,38,40,41,42,43]. In addition, exosome therapy was shown to contribute to neuronal preservation and to have neuroprotective and regenerative effects on neurons, synapses, and myelin sheaths, as demonstrated in models of neuroinflammation, Alzheimer’s, stroke, and SCI [37,39,42,44,45,46,47,48]. The ability of exosomes to prevent scar tissue formation also contributed to regeneration in SCI [39].

In addition to the general therapeutic effects reported after treatment with MSC-derived exosomes, benefits specific to neurodegenerative diseases were also observed. For example, exosomes could restore cognition impairment and rescue CA1 synaptic transmission and long-term potentiation (LTP) in mouse models for AD where memory loss is a major symptom [46,49,50,51]. Moreover, MSC-derived exosomes ameliorated the destructive structural changes in the taste buds and their innervations, which is also a manifestation of AD [52]. In stroke, treatment with MSC-derived exosomes enhanced recovery of fine motor function, improved spatial learning and memory ability, reduced the neurological severity score, and reduced infarct size [33,34,36,40,47,53,54,55]. In perinatal brain injury, exosome therapy also improved long-term neurodevelopmental outcome cognitive functions, and reduced the total number and duration of seizures, characterizing the disorder [43,56,57,58]. Further benefits that have been attributed to MSC-derived exosome treatment include cognitive and sensorimotor improvement, reduced spatial learning impairments, and reduced cortical lesion volume in TBI [41,59,60,61]; stimulation of locomotor functional recovery and improved mechanical sensitivity in SCI [42,48,62,63]; and ameliorated learning, cognitive and memory impairments in SE [64,65].

Studies on the therapeutic potential of MSC-derived exosomes have used both rodent and human-derived exosomes (as listed in Table 1 and Table 2, respectively), since both sources appear to yield promising results. Although the most popular route of administration is intravenous (IV) injection, it was demonstrated, using an in vivo neuroimaging, that MSC-derived exosomes can cross the blood–brain barrier (BBB) after intranasal administration more efficiently compares to IV injection [66]. This was further supported in near-infrared (NIR) imaging also showed that the intranasal administration delivered DiR-labeled MSC-derived exosomes into the brain, whereas tail vein injection primarily resulted in liver and kidney [67]. Nevertheless, no comparative study was performed to analyze all route of administration in the same model in order to examine functional efficiency.

Exosomes from a variety of different tissues demonstrated promising results as a therapy for neurodegenerative diseases in animal models. However, only a few comparisons of the different sources of MSC-derived exosomes were performed in general, and in the context of neurodegeneration in particular. Tracy et al. showed that both amniotic fluid MSCs (afMSCs) and bone marrow (BM) MSCs can provide exosomes with similar morphology, size distributions, and expression of tetraspanin markers [68]. Nevertheless, afMSCs seem to produce more exosomes per cell under the same culture conditions. When compared exosome fractions of human menstrual (MenSCs), BM, umbilical cord, and chorion MSC, MenSC exosomes showed superior effects on the growth of the longest neurite in cortical neurons and had a comparable effect to BM-MSC exosomes on neurite outgrowth in dorsal root ganglia neurons [69]. It appears that no proper comparison between exosomes from different MSC sources was conducted in terms of functional outcome in neurodegenerative animal models, and therefore it is not possible to predict whether the efficacy of exosomes from different sources is comparable. Further research is, therefore, needed to determine whether differences between them exist. Further research is also needed in Parkinson’s disease, which is noticeable by its absence from the list of neurodegenerative diseases investigated in the context of MSC-derived exosome therapy.

## 4. Mechanism of Action

The efficacy of treatment with MSC-derived exosomes, has been explained by their ability to remove, or inhibit pathological processes on one hand, and by promotion of regenerative mechanisms on the other (Figure 1). Such activities include reduction of amyloid beta (Aβ) aggregates in AD [45,46,51], reduction of demyelination in MS [28,29], and inhibition of apoptosis, as observed in stroke, TBI, and SCI [31,32,38,39,40,41,42]. Immunomodulation, including inhibition of secretion of pro-inflammatory cytokines, together with an induction of anti-inflammatory factors, was observed in all neurodegenerative diseases.

In the context of regeneration, there are four major mechanisms through which MSC-derived exosomes achieve the improved outcomes described above, are neuroprotection, neurogenesis, neuromodulation, and angiogenesis. Neuroprotection is a broad term referring to the prevention of cell death and the restoration of neuronal numbers, as well as the functional restoration of damaged neurons [71]. MSC-derived exosomes were observed to exert neuroprotection in models of AD by the reduction of dystrophic neurites [45], in stroke by increasing the connectivity and remodeling of neurites [32,37,70], in SCI by rehabilitation of axons and synapses [39,42,48], and in SE by reducing glutamatergic and GABAergic neuronal loss [65]. Neurogenesis following MSC-derived exosome treatment was reported in Alzheimer’s, stroke, and TBI [37,50,54,59,70,72]. Immunomodulatory processes including modulation of microglial activity, stimulation of regulatory T cells, modulation of the polarization state of microglia/macrophages, and inhibition of reactive astrocytes, were manifested in AD, MS, stroke, perinatal brain injury, and TBI [28,29,33,36,37,40,41,43,51,52,57,59]. Generation of new blood vessels has mainly been reported in neurodegenerative diseases involving the vascular system, namely, stroke, TBI, and SCI [31,36,37,54,59,70,72].

Besides these four major mechanisms, MSC-derived exosome treatment may also provide protection from insults by reducing oxidative stress [36,44] and restoring the integrity of the BBB [40]. Since the etiology of neurodegenerative diseases is complex, the mechanisms by which recovery can occur are complex and intertwined. The ability of MSC-derived exosomes mechanisms to promote regeneration is not unexpected since the exosomes reflect the cells from which they are derived, and therefore have similar properties, including the mechanisms of action [4].

The exact molecular mechanism of action through which MSC-derived exosomes operate is not fully understood due to the wide variety of molecules comprising the exosomal cargo. ExoCarta, an exosome database contains 41,860 protein, 1116 lipid molecule, 3408 mRNA, 2838 miRNA entries, derived from studies of exosomes in several species [73]. Thereby, a variety of functions and multiple molecules can be excreted from exosomal cargo [74]. Among other functions, exosomal proteins can act as signaling molecules, receptors, cell adhesion molecules. For example, the expression of proteins such as nestin, neuro-D, growth-associated protein 43, synaptophysins, VEGF, FGF promote events such as neural development, synaptogenesis, and angiogenesis [75]. Moreover, it was indicated that MSC-derived exosomes from adipose tissue contain neprilysin, an enzyme capable of degrading Aβ, and in co-culture with cells designed for Aβ exacerbated production, these exosomes significantly reduced levels of Aβ1–40 and 1–42 [76]. Furthermore, MSC-Exo contain several immunomodulatory factors including transforming growth factor-β (TGF-β), hepatic growth factor (HGF), indolamine 2,3-dioxygenase-1 (IDO-1), interleukin (IL)-10, IL-1 receptor antagonist (IL-1Ra), and prostaglandin E2 (PGE2) [26].

miRNA can also control functions related to neural remodeling as well as angiogenic and neurogenic processes [74]. It has been reported that exosomes also contain miR-98, miR-155, and miR-125a, which have antiapoptotic activity [77,78]. miR-143 and miR-21, which play an important role in immune response modulation and in neuronal death associated with an environment of chronic inflammation were also found to be present in MSCs-derived exosomes [79]. Similarly, a miRNA cluster formed by miR-17, miR-18a, miR-19a/b, miR-20a, and miR-90a, was also found to be present in MSCs-derived exosomes, and described as important modulators of neurite remodeling and neurogenesis, as well as stimulators of axonal growth and CNS recovery [80]. Overall, there are multiple potential pathways through which MSC-derived exosomes may operate. Nevertheless, as mentioned above, there are several suggested pathways that has been implicated in the exosomal mechanism of action.

## 5. Limitations of Current Knowledge

Despite the great beneficial effect of exosomes in preclinical trials, there remain a number of unresolved issues that need to be addressed before their use in clinical therapy. Despite the short-term survival of infused MSCs, their beneficial effects have been demonstrated to persist over time in variety of disease models, even when there is no evidence of their continued presence [8,9,15,16].

The half-life of exosomes in vivo is estimated to be minutes, and most exosomes have been shown to evacuate within a few hours [81,82,83]. Exosomes circulate in the blood, and transport their cargo into the target cell via fusion, receptor-mediated endocytosis, micropinocytosis, or phagocytosis [19,84]. That is, exosomes in their original form are rapidly cleared from the body. There is the possibility that the contents of the exosomes mediate activation of a cascade whose effect is maintained over time. Nevertheless, this possibility cannot be simply assumed, and the question of whether exosome treatment is likely to have a long-lasting effect requires further investigation.

Exosome treatment holds great therapeutic promise, even if repeated treatments prove to be needed. In that case, it will be necessary to scale up exosome production in a repeatable manner, which in itself may pose a difficulty. There are three important issues that are prevalent in good manufacturing practice (GMP) for exosomes: upstream of cell cultivation process, downstream of the purification process, and exosome quality control [85]. Because exosomes are secreted by cells, a production system could be established using a large-scale cell cultivation system. A hollow fiber-based bioreactor for cell culture is an attractive strategy for exosome production because of the advantage that decreased volume of condition medium can be harvest from the filtrated fiber. The downstream purification system should preferably conform to the procedures of vaccine production because of the similarity in particle size and features of secretory vesicles of the host cells. Exosomes purified by ultrafiltration for avoiding bioactive protein release from vesicles of exosomes were shown to have a higher benefit than those of ultracentrifugation. The challenge in GMP of exosomes is quality control. Although markers of exosomes have been defined by previous studies, the type of cells producing exosomes is diverse. The determination of biofunctions, such as biomarker of exosomes and properties derived from parental cells, are the two major issues for characterization of exosomes before application in clinical trials. In addition, to scale up production, it will also be necessary to ensure that repeated treatments do not elicit an undesirable immune response.

It is also important to note that despite the remarkable functional outcomes and improvements achieved in the animal models, recovery is usually incomplete. Therefore, there is a great value in research designed to further improve MSC-derived exosome treatment.

## 6. Toward MSC-Derived Exosome-Based Therapies

Apart from their therapeutic capability, MSC-derived exosomes have also been shown to have the ability to migrate to lesion sites. This feature is extremely important, especially when treating neurodegenerative diseases, as the ability to reach the brain is extremely limited. Using an in vivo exosome neuroimaging technique, intranasally-administered MSC-derived exosomes were shown to specifically target sites of brain lesions generated in various pathological murine models [86,87]. Exosomes accumulated in lesions up to 96 h post-treatment, although they showed a diffuse migration pattern and clearance by 24 h post-delivery in healthy controls. The importance of this feature lies in the possibility of using exosomes not only as an independent therapy, but also as a delivery system that can transport drugs directly to the lesion site.

Studies designed to exploit this approach have introduced potentially therapeutic molecular agents into MSC-derived exosomes (Table 3). The most common type of molecular agent used for this purpose is miRNA, which can be used to supplement a deficiency that exists in a particular pathology. The miRNAs miR-29b-3p, miR-126, and miR-30d-5p, which were all found to be downregulated after ischemic injury, were therefore inserted into MSC-derived exosomes for the treatment of stroke [88,89,90]. The results indicated an attenuation of brain injury, that was greater than when naïve exosomes were used.

The use of miRNA is also designed to exploit inherent properties that may contribute to mitigating damage caused by central nervous system (CNS) disorders. For example, miR-124, is known to regulate the function of microglia under physiological conditions [91]. When internalized in MSC-derived exosomes, miR-124 improved neurological function recovery in rat models of TBI [91]. Similarly, miR-210, miR-17-92, and miR-133b were loaded into MSC-derived exosomes as stroke treatments [92,93,94] based on the observations that miRNA-210 promotes angiogenesis [94]; miR-17-92 increases cell proliferation, inhibits cell death and contributes to axonal outgrowth [93]; while miR-133b regulates the production of tyrosine hydroxylase and dopamine receptors and promotes the outgrowth of neurites [92,95]. The results indicated an enhanced survival rate [94], improved neurological outcome [93], and a reduction in apoptotic and neurodegenerative neurons [92], respectively.

In addition, miR-133b exosomes were shown to improve recovery of hindlimb locomotor function in an SCI rat model [95]. miR-25 and miR-29b also enhanced exosomal function in rodent models of SCI [96,97]. miR-25, which is known to promote neural stem cell proliferation, inhibit cell apoptosis, and regulate oxidative stress, was shown to improve motor deficit index (MDI) and enhance neuroprotection when introduced inside MSC-derived exosomes [96]. Furthermore, miR-29b, which was shown to be involved in the repair of liver damage, myocardial ischemia-reperfusion injury, skeletal muscle injury, as well as human podocyte injury, was able to increase BBB score in SCI, when contained in MSC-derived exosomes [97]. Another agent used is the phosphatase and tensin homolog (PTEN) siRNA [98]. PTEN is expressed in neurons and regenerating axons and plays a vital role in controlling the regeneration of corticospinal neurons. For this reason, PTEN siRNA, which, like miRNA, is a small RNA that regulates gene expression, has been inserted into MSC-derived exosomes and used in a rat model of spinal cord injury, where it elicited functional recovery and improved structural and electrophysiological function [98].

miRNAs are suitable for insertion into exosomes because of their small size, and their ability to generate a cascade of events. Nevertheless, miRNAs are not the only molecular agents that can be inserted into MSC-derived exosomes. LJM-3064 aptamer combined with MSC-derived exosomes were shown to reduce the areas of demyelination and ameliorate disease severity in an experimental autoimmune encephalomyelitis (EAE) mouse model of MS [99]. Both enkephalin and pigment epithelium-derived factor (PEDF) were loaded into exosomes and introduced into an MCAo rat model of focal stroke, resulting in improved brain neuron density and neurological score [100]; and a reduction in infarct volume and neuronal apoptosis [101], respectively. Similarly, MSC-derived exosome loaded with curcumin suppressed cellular apoptosis in the lesion region in a mouse model of stroke [102]. Moreover, MSC-derived exosomes enriched with BDNF were found to inhibit apoptosis and promote neuronal regeneration in a TBI rat model [103].

Functionality in vivo can be enhanced by loading exosomes with molecular agents that improve the migratory capacity and thus increase the number that reach the site of damage. In this context, the rabies viral glycoprotein (RVG) peptide was shown to interact specifically with the acetylcholine receptor, making it specific to the CNS [104]. Modifying exosomes to include RVG, enhanced the engraftment of exosomes in the cortex and hippocampus of AD brains. Thus, they significantly improved learning and memory function [104]. In an analogous fashion, magnetic nanovesicles (MNV) derived from MSCs that had internalized iron oxide nanoparticles (IONP) were shown to have dramatically improved targeting to the ischemic-lesion and superior the therapeutic outcomes [105]. Due to the high expression of the transferrin receptor after stroke, transferrin has also been used as a target agent in order to transfer exosomes to ischemic brains [100]. The migration ability of the transferrin loaded exosomes (tar-exo) was higher than that of naïve exosomes leading to improvements in neurological recovery [100]. Furthermore, exosomes can be modified with the cyclo(Arg-Gly-Asp-D-Tyr-Lys) peptide [c(RGDyK)] using bio-orthogonal copper-free azide alkyne cyclo-addition (click chemistry) [94,102]. This peptide exhibits a high affinity to integrin αvβ3. Following conjugation to the surface of MSC-derived exosomes, they specifically targeted reactive cerebral vascular endothelial cells, following ischemia. This was demonstrated when engineered exosomes targeted the lesion region of ischemic brains following IV administration [94,102].

Insertion of molecular agents into exosomes can be accomplished by a variety of methods, either directly into the exosomes, or into the cells from which the exosomes are derived. Direct methods include chemical reactions [94,99,102,104], electroporation [100], cholesterol-conjugation (hydrophobic reaction) [94,98], and incubation [102]. Indirect methods targeting the cells include transfection (via lipofectamine or virus infection) [89,91,92,95,96,97,101], electroporation [106], or addition as a media supplement [103,105]. The efficiency of these methods has not yet been compared and, therefore, no information is available regarding the optimal loading method. However, there is evidence from literature reports that the efficiency of naïve MSC-derived exosome therapy can be further enhanced by loading a variety of molecular agents, and using the exosomes as a delivery system. Such regimens may enable the use of lower doses of medications, and minimize adverse effects resulting from systemic treatment.

## 7. MSC-Derived Exosomes in Clinical Trials

Over 200 clinical trials of exosomes or extracellular vesicles treatments are listed on the clinicaltrials.gov website. Nine of these studies use MSC-derived exosomes. Of the clinical studies that use MSC-derived exosomes, only one study is relevant to this review—Allogenic Mesenchymal Stem Cell-Derived Exosome in Patients with Acute Ischemic Stroke (NCT03384433).

This study was conducted by Prof. Alireza Zali from the Shahid Beheshti University of Medical Sciences and examines the safety and efficacy of MSC-derived exosomes enriched with miR-124, in acute ischemic stroke patients. The study was based on a preclinical study that demonstrated the ability of miR-124 loaded MSC-derived exosomes to relieve brain injury by promoting neurogenesis [91]. As part of the clinical study, stereotaxis was used to deliver MSC-derived exosomes loaded with miR-124 (200 mg protein), to the ischemic stroke area of five stroke patients, one month after the stroke. The primary outcome measure of the study was safety, i.e., documenting adverse incidents, including deteriorating stroke, stroke recurrences, brain edema, seizures, and hemorrhagic transformation during the 12 months following treatment. The secondary outcome measure was efficacy, i.e., measuring the degree of disability of the patients using the modified Rankin scale during the first year after treatment. The study was completed in December 2019, but the clinical findings have not yet been published.

Despite impressive preclinical results in both clinical and biochemical parameters, the use of MSC-derived exosomes in clinical trials is still limited. This is mainly due to the necessity to transition to robust large-scale production. We predict that once this obstacle is overcome, and if positive results are obtained in existing clinical trials, further studies will be carried out on patients with various diseases, and in particular in neurodegenerative diseases.

## 8. Summary

Upon administration, MSC-derived exosomes can specifically target and accumulate in brain lesion sites in various murine models of diseases, where they improve the behavioral phenotype as well as reducing the inflammatory response. This improvement is not inferior to that obtained following the transplantation of the parent cells and has the advantage of avoiding adverse events associated with whole cell transplants. The exosomes orchestrate a series of events that enable recovery and regeneration in neurodegenerative diseases. Advanced studies have attempted to exploit and improve the homing feature of MSC-derived exosomes in order to deliver molecular agents to brain lesions and enhance recovery. Combining the intrinsic properties of exosomes with a targeted medication is suggested as a novel therapeutic approach that might have a dramatic impact on the future of neurodegenerative disease therapy.

## Figures and Tables

**Figure 1 biomolecules-10-01320-f001:**
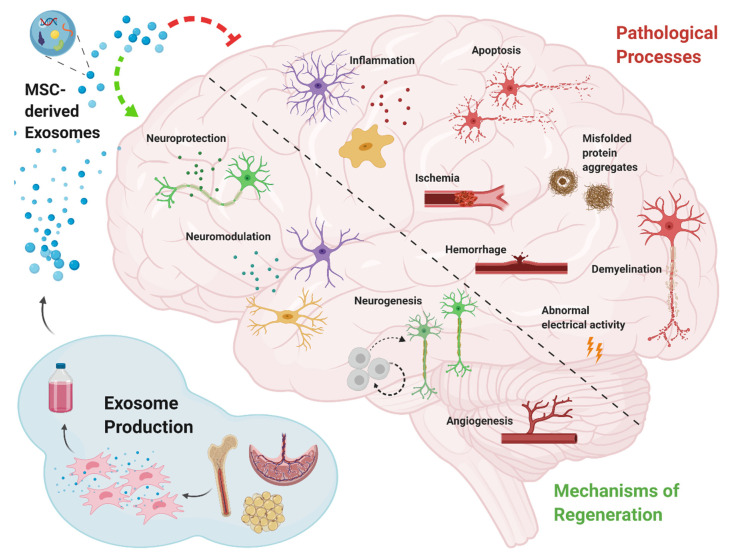
Schematic diagram depicting the major pathological processes in neurodegenerative diseases, and the key mechanisms through which mesenchymal stem cell-derived exosomes operate in order to mitigate these pathologies and induce regeneration.

**Table 1 biomolecules-10-01320-t001:** Naïve non-human MSC-derived exosomes in animal models of neurodegeneration.

Disease/Disorder	Reference	Animal Model	Cell Source	Dose	Route of Administration	Biological/Medical Improvement	Suggested Mechanism of Action
Alzheimer’s	[52]	Ovariectomized albino-rat	Rat BM	100 μg	Intravenous (IV)	Improved in destructive structural changes in the taste buds and their innervations	Improved synaptophysin-immunoreactivity
Alzheimer’s	[45]	APP/PS1 mouse	Mouse BM	22.4 µg	Intracerebral	Reduced amount of dystrophic neurites in both the cortex and hippocampus	Aβ plaque reduction
Alzheimer’s	[49]	Streptozotocin-induced mouse	Mouse BM	0.5 μg/day for 5 days	Intraventricular	Recovered cognition impairment	Not mentioned
Alzheimer’s	[46]	APP/PS1 mouse	Mouse BM	100 μg	Intracerebroventricular	Improved cognitive behavior, rescued impairment of CA1 synaptic transmission, and long-term potentiation	Suppression of Aβ induced iNOS expression
MS	[28]	EAE rat	Rat BM	100/400 μg	IV	Decreased neural behavioral scores	Reduced demyelination and neuroinflammation
Stroke	[32]	Subcortical infarction rat	Rat adipose	50/100/200 μg	IV	Improved functional outcomes associated with decreased cell death	Restored fiber tract connectivity, increased oligodendrocyte markers, and re-myelination
Stroke	[33]	MCAo rat	Rat BM	120.68 μg	IV	Reduced neurological severity score; improved spatial learning and memory ability	Inhibited the expression of CysLT2R and NMLTC4 treated microglia; modulated the balance between M1 and M2 microglia; decreased pro-inflammatory cytokines secretion; increased anti-inflammatory and neurotrophic factors production
Stroke	[53]	Cortical injured monkey	Monkey BM	4 × 10^11^ particles/kg	IV	Enhanced recovery of fine motor function	Not mentioned
Stroke	[40]	MCAo rat	Adipose (cell source not mentioned)	3 treatments of 2.0 × 10^6^ particles	IV	Reduced infarct volume; suppressed apoptosis	Improved BBB condition; suppressed inflammation; reduced abnormal high level of miR-21-3p
Stroke	[54]	Intracerebral hemorrhage injection rat	Rat BM	100 µg	IV	Improved spatial learning, motor function, and sensory memory	Promoting endogenous angiogenesis and neurogenesis; increased white matter remodeling
Stroke	[70]	tMCAo rat	Rat BM	30 µg	IV	Improved Neurological function	Promoted neurogenesis and angiogenesis via miR-184 and miR-210, respectively
Stroke	[34]	Intracerebral hemorrhage rat	Rat adipose	100 µg	IV	Improved functional recovery; reduced infarct size	Increased fiber tract and axonal sprouting; enhanced oligodendrocyte formation and remyelination
Stroke	[47]	Transient global cerebral ischemia mouse	Mouse BM	200 µg	Intracerebroventricular	Restored impaired basal synaptic transmission and synaptic plasticity, and improved spatial learning and memory	Inhibited pathogenic expression of COX-2 in the hippocampus
Stroke	[35]	Subcortical infarct rat	Rat adipose	100 µg	IV	Improved functional recovery	Increased axonal sprouting and growth, oligodendrocyte formation, tract connectivity and remyelination
Stroke	[36]	MCAo rat	Mini-pig adipose	100 µg	IV	Reduced brain infarct zone; improved neurological function	Suppressed inflammation; reduced ROS and oxidative stress generation; promoted angiogenesis
Stroke	[69]	MCAo rat	Rat BM	100 µg	IV	Improved neurologic outcome	Enhanced neurite remodeling, neurogenesis, and angiogenesis
Neuroinflammation	[44]	LPS-induced rat	Rat BM	200 µg	IV	Enhanced neuronal survival	Reduced oxidative stress; reduced inflammatory response
TBI	[41]	Controlled cortical impact (CCI) mouse	Rat BM	30 µg	Retro-orbital	Improved functional recovery; reduced cortical lesion volume; attenuated cellular apoptosis	Inhibited early neuroinflammation through modulation of microglia/macrophages polarization
TBI	[59]	CCI rat	Rat BM	100 µg	IV	Cognitive and sensorimotor improvement.	Promotion of endogenous angiogenesis and neurogenesis; and inflammation reduction.
SCI	[38]	Spinal cord hemisection rat	Rat BM	100 µg	IV	Improved functional recovery and attenuated lesion size and apoptosis	Targeted inhibition of the FasL gene by miR-21-5p
SCI	[42]	Rat contusive SCI	Rat BM	1 × 10^6^ cells equivalents	IV	Reduced neuronal cell apoptosis, enhanced neuronal survival and regeneration, and improved motor function	Suppression of pericytes migration; and improved blood-spinal cord barrier integrity via NF-κB p65 signaling
SCI	[39]	Rat contusive SCI	Rat BM	100 µg	IV	Suppressed glial scar formation; attenuated lesion size; promoted axonal regeneration; and improved functional behavioral recovery	Promoted blood vessel formation; reduced neuronal cells apoptosis; suppressed inflammation; and suppressed activation of A1 neurotoxic reactive astrocytes
SCI	[30]	Spinal cord hemisection injured rat	Rat BM	100 µg	IV	Reduced disease severity	Inhibited complement mRNA synthesis and release; inhibited activation of NF-κB signaling by binding to microglia cells.
SCI	[48]	Rat contusive SCI	Rat BM	1 × 10^6^ cells equivalents	IV	Improved locomotor function; and the neuroprotective effect on residual neurons, synapses, and myelin sheath.	Reduced A1 astrocyte proportion by inhibiting NFκB activation; reductions in proinflammatory cytokine levels
SCI	[31]	Rat contusive SCI	Rat BM	100 µg	IV	Attenuated lesion size and improved functional recovery	Attenuated cellular apoptosis and inflammation; promoted angiogenesis

**Table 2 biomolecules-10-01320-t002:** Naïve human MSC-derived exosomes in animal models of neurodegeneration.

Disease/Disorder	Reference	Animal Model	Cell Source	Dose	Root of Administration	Biological/Medical Improvement	Suggested Mechanism of Action
Alzheimer’s	[50]	Aβ-inoculated mouse	Human, purchased from ATCC	10 µg	Intrahippocampal	Enhance neurogenesis and restore cognitive function	Not mentioned
Alzheimer’s	[51]	APP/PS1 mouse	Human umbilical cord	30 µg	IV	Repair cognitive disfunctions	Help to clear Aβ deposition; and modulate the activation of microglia in the brain
MS	[29]	EAE mouse	Human BM	150 μg	IV	Reduced disease severity	Reduced demyelination; decreased neuroinflammation; and upregulated the number of regulatory T cells
Stroke	[55]	MCAo rat	Human umbilical cord blood	150 µg	IV	Attenuated infarct size; exacerbated the somatosensory and motor dysfunction	Not mentioned
Stroke	[37]	MCAo mouse	Human BM	Released by 2 × 10^6^ MSCs	IV	Improved neurological impairment and long-term neuroprotection	Promoted neurogenesis and angiogenesis; prevented post-ischemic immunosuppression
Perinatal brain injury	[56]	A combination of a hypoxic-ischemic and an inflammatory insult in rat	Human Wharton’s jelly	50 mg/kg	Intranasal (IN)	Improved long-term neurodevelopmental outcome	Prevented gray and white matter alterations
Perinatal brain injury	[43]	Rice-Vannucci mouse	Human BM	1.25 × 10^9^ particles/dose	IN	Improved short-term behavioral outcomes; reduced tissue volume loss and cell death	Reduced microglial activation
Perinatal brain injury	[57]	LPS-induced rat	Human BM	1 × 10^8^ cell equivalents/kg bodyweight	Intraperitoneal (IP)	Improved long-lasting cognitive functions	Ameliorated inflammation-induced neuronal cellular degeneration; reduced microgliosis; prevented reactive astrogliosis; and restored short-term myelination deficits and long-term microstructural abnormalities of the white matter
Perinatal brain injury	[58]	Transient umbilical cord occlusion in preterm ovine fetus	Human BM	Two boluses of 2.0 × 10^7^ cell equivalents	IV	Reduced total number and duration of seizures; and preserved baroreceptor reflex sensitivity	Hypomyelination prevention
TBI	[61]	A combination of CCI and hemorrhagic shock swine	Human BM	1 × 10^13^ particles	IV	Reduced the severity of neurological injury and improved neurocognitive recovery	Not mentioned
TBI	[60]	CCI mouse	Human BM	30 µg	IV	Rescued pattern separation and spatial learning impairments	Immunomodulation
SCI	[62]	Mouse contusive SCI	Human umbilical cord	20/200 µg	IV	Promoted locomotor functional recovery	Attenuated inflammation of the injury region
SCI	[63]	Spinal cord contusion rat	Human BM	1 × 10^9^ particles	IV	Improved locomotor recovery score; improved mechanical sensitivity	Diminished inflammatory response
SE	[64]	Pilocarpine mouse	Human umbilical cord	30 µg	Intraventricular	Ameliorated learning and memory impairments	Reduced inflammatory responses associated with hippocampal astrocyte activation via Nrf2-NF-κB signaling pathway
SE	[65]	Pilocarpine mouse	Human BM	30 µg	IN	Long-term preservation of normal hippocampal neurogenesis and cognitive and memory function	Diminished loss of glutamatergic and GABAergic neurons; and reduced inflammation in the hippocampus

**Table 3 biomolecules-10-01320-t003:** Enriched mesenchymal stem cell (MSC)-derived exosomes in animal models of neurodegeneration.

Disease/Disorder	Reference	Animal Model	Cell Source	The Addition	Dose	Route of Administration	Biological/Medical Improvement	Suggested Mechanism of Action
Alzheimer’s	[92]	APP/PS1 mouse	Mouse BM	Rabies viral glycoprotein (RVG)	4 boluses of 5 × 10^11^ particles	IV	Improved learning and memory function	Decreased plaque deposition and Aβ levels; reduced astrocytes activation; reduced pro-inflammatory mediators and raised anti-inflammatory factors
MS	[87]	EAE mouse	Mouse BM	LJM-3064 aptamer	200 μg	IV	Reduced disease severity	Suppressed of inflammatory response; lowered demyelination lesion region
Stroke	[77]	MCAo rat	Rat BM	miR-29b-3p	100 μg/kg/day for 3 days	Intracerebroventricular	Reduced infarct volume	Suppressed neuronal apoptosis and promoted angiogenesis through the downregulation of PTEN and activation of Akt signaling pathway
Stroke	[93]	MCAo rat	Human BM	Iron oxide nanoparticles (IONP)	200 μg	IV	Decreased infarction volume and improved motor function	Promoted the anti-inflammatory response, angiogenesis, and anti-apoptosis
Stroke	[78]	MCAo rat	Rat adipose	miR-126	Not mentioned	IV	Enhanced functional recovery	Inhibited microglial activation and inflammatory response; promoted neurogenesis and vasculogenesis
Stroke	[88]	Transient MCAo rat	Rat BM	Transferrin and enkephalin	One or two boluses of 5 × 10^4^	IV	Improved brain neuron density and neurological score	Decreased levels of LDH, p53, caspase-3, and NO
Stroke	[82]	MCAo rat	Mouse BM	c(RGDyK) peptide and miR-210	100 µg	IV	Enhanced survival rate	Promoted angiogenesis; up-regulation of integrin β3 and CD34 expression
Stroke	[90]	MCAo mice	Mouse BM	c(RGDyK) peptide and curcumin	100 µg	IV	Reduced cellular apoptosis in the legion region	Suppressed inflammatory response
Stroke	[15]	Intracerebral hemorrhage rat	Rat BM	miR-133b	100 μg	IV	Reduced apoptotic and neurodegenerative neurons	Inhibited RhoA and activation of ERK1/2/CREB pathway
Stroke	[89]	MCAo rat	Rat adipose	Pigment epithelium-derived factor (PEDF)	100 μg/kg/day for 3 days	Lateral cerebral ventricle	Reduced infarct volume; suppressed neuronal apoptosis	Activated autophagy
Stroke	[79]	Modified MCAo rat	Rat adipose	miR-30d-5p	80 μg	IV	Decreased cerebral injury area of infarction	Suppressed autophagy and promoted M2 microglia/macrophage polarization
Stroke	[81]	MCAo rat	Rat BM	miR-17-92	100 µg	IV	Improved neurological outcome	Increased neural remodeling including neurogenesis, oligodendrogenesis and neurite plasticity; inhibited PTEN, and subsequently increased the phosphorylation of PTEN downstream proteins, Akt, mTOR and GSK-3β
TBI	[91]	Electric cortical contusion impactor rat	Rat BM	BDNF	100 µg	IV	Inhibit apoptosis	Inhibited inflammation and promoted neuronal regeneration; increased miR-216a-5p
TBI	[80]	Controlled cortex injury rat	Rat BM	miR-124	100 µg	IV	Improved neurological function recovery	Reduced production of pro-inflammatory cytokines; promoted M2 polarization of microglia; increased production of anti-inflammatory cytokines; enhanced neurogenesis in hippocampus
SCI	[86]	Complete spinal cord transection rat	Human BM	Phosphatase and tensin homolog (PTEN) siRNA	5 boluses of 1.62 × 10^8^ particles	IN	Elicited functional recovery; improved structural and electrophysiological function	Enhanced axonal growth and neovascularization; reduced microgliosis and astrogliosis
SCI	[84]	Spinal cord ischemia rat	Rat BM	miR-25	20 µg	Intrathecal	Improved MDI (motor deficit index); enhanced neuroprotection	Reduced pro-inflammatory cytokines; reduced oxidative stress markers
SCI	[85]	Rat contusive SCI	Rat BM	miR-29b	100 µg	IV	Increased BBB score	Decreased contractile nerve cell numbers and GFAP positive neurons
SCI	[83]	Compression SCI rat	Rat BM	miR-133b	100 μg	IV	Improved recovery of hindlimb locomotor function	Preserved neurons; promoted regeneration of axons; activated ERK1/2, STAT3, and CREB; inhibited RhoA expression

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
