# Peer review of "Promising Opportunities for Treating Neurodegenerative Diseases with Mesenchymal Stem Cell-Derived Exosomes"

_biomolecules, 2020, doi:10.3390/biom10091320_

Round 1

Reviewer 1 Report

In the present manuscript, the authors challenge to give an overview of the potentialities of MSCs-derived exosomes as a promising therapeutic tool and route to central nervous system neurodegenerative diseases. The review is concise and appropriate, tackling one of the big issues of a field that is currently under a remarkable investigation, also addressing its strong points and limitations.

Notwithstanding, there are some issues that should be considered, namely:

  1. In the tables presented, the authors have demonstrated promising results from MSCs-derived exosomes from different sources. Therefore, does the source matter? Or it depends on the model/disease used? This should be considered and addressed in a section, as there are indications that the paracrine action of MSCs from different sources have a different secretory profile, as well as a different impact according to the model/disease to be used. 
  2. The authors have addressed just the potential role o MSCs-derived exosomes, but any idea about the comparison of MSCs-derived exosomes and MSCs-derived secretome (as a whole)?
  3. In line 163, the authors say "it will be necessary to scale up exosome production in a repeatable manner". How? Bioprocessing? Bioreactors? 3D cultures? This should be clearly stated in the manuscript.

Author Response

  1. In the tables presented, the authors have demonstrated promising results from MSCs-derived exosomes from different sources. Therefore, does the source matter? Or itdependson the model/disease used? This should be considered and addressed in a section, as there are indications that the paracrine action of MSCs from different sources have a different secretory profile, as well as a different impact according to the model/disease to be used. 

Answer:

Thank you very much for you comment.

As described previously, MSC-derived exosomes from different sources demonstrated promising results as a therapy for neurodegenerative diseases in animal models. However, only few comparisons of the different sources of MSC-derived exosomes were performed in general, and in the context of neurodegeneration in particular. Tracy et al. showed that both amnionic fluid MSCs (afMSCs) and bone marrow (BM) MSCs can provide exosomes with similar morphology, size distributions, and expression of tetraspanin markers [1]. Nevertheless, afMSCs seem to produce more exosomes per cell under the same culture conditions. When compared exosome fractions of human menstrual (MenSCs), BM, umbilical cord and chorion MSC, MenSC exosomes showed superior effects on the growth of the longest neurite in cortical neurons and had a comparable effect to BM-MSC exosomes on neurite outgrowth in dorsal root ganglia neurons [2]. It appears that no proper comparison between exosomes from different MSC sources was conducted, in terms of functional outcome in neurodegenerative animal models, and therefore it is not possible to predict whether the efficacy of exosomes from different sources is comparable. Further research is therefore needed to determine whether differences between them exist.

This section was inserted within the text (highlighted).

  1. The authors have addressed just the potential role o MSCs-derivedexosomes, but any idea about the comparison of MSCs-derived exosomes and MSCs-derived secretome (as a whole)?

Answer:
This is an interesting note.

MSC-derived secretome is defined as the set of MSC-derived bioactive factors (soluble proteins, nucleic acids, lipids and extracellular vesicles (EVs)) secreted to the extracellular space [3]. The characterization of BM-MSC-conditioned media by antibody-based protein array analyses and enzyme-linked immunosorbent assays (ELISAs) showed enrichment of IGF-1, HGF, VEGF and TGF-b, and provided evidence of their role in the increased neuronal survival and outgrowth in vitro. Human BM-MSC (hBMSCs) secretome was also shown to induce higher levels of in vitro neuronal differentiation [4]. Moreover, various biological effects were observed in experimental animals after administration of MSC-CM [5]. Both EVs and soluble component of MSC-CM were capable to promote tissue regeneration, suppress detrimental immune response and induce neo-angiogenesis in ischemic tissues [6], [7]. The injection of hBMSCs secretome also led to the rescue of DA neurons in a rat PD model, when compared to transplantation of hBMSCs [4]. MSC-sourced secretomes showed immunoregulatory, angiomodulatory and anti-apoptotic effects that resulted in enhanced tissue repair and regeneration [5]. It can be therefore concluded that MSCs-derived exosomes and MSCs-derived secretome have very similar properties. Nevertheless, exosomes have the advantage of having a membranous shell that preserves their contents, in contrast to free factors in the secretome which are unprotected and prone to degradation. In addition, the exosomal membrane enables the migratory capability of the exosomes, a feature that does not exist in the secretome as a whole.

This information was not included in the text as it deviates from the main topic of the paper and is therefore found to be less relevant. If the reviewer finds it appropriate to include the topic in the text, it will be added.

  1. In line 163, the authors say "it will be necessary to scale up exosome production in a repeatable manner". How? Bioprocessing? Bioreactors? 3D cultures? This should be clearly stated in the manuscript.

Answer:

There are three important issues that are prevalent in good manufacturing practice (GMP) for exosomes: upstream of cell cultivation process, downstream of the purification process, and exosome quality control [8]. Because exosomes are secreted by cells, a production system could be established using a large-scale cell cultivation system. A hollow fiber-based bioreactor for cell culture is an attractive strategy for exosome production because of the advantage that decreased volume of condition medium can be harvest from the filtrated fiber. The downstream purification system should preferably conform to the procedures of vaccine production because of the similarity in particle size and features of secretory vesicles of the host cells. Exosomes purified by ultrafiltration for avoiding bioactive protein release from vesicle of exosomes were shown to have higher benefit than those of ultracentrifugation. The challenge in GMP of exosomes is quality control. Although markers of exosomes have been defined by previous studies, the type of cells producing exosomes is diverse. The determination of biofunctions, such as biomarker of exosomes and properties derived from parental cells, are the two major issues for characterization of exosomes before application in clinical trials.

This section was inserted within the text (highlighted).

[1]        S. A. Tracy et al., “A comparison of clinically relevant sources of mesenchymal stem cell-derived exosomes: Bone marrow and amniotic fluid,” J. Pediatr. Surg., vol. 54, no. 1, pp. 86–90, Jan. 2019.

[2]        M. A. Lopez-Verrilli, A. Caviedes, A. Cabrera, S. Sandoval, U. Wyneken, and M. Khoury, “Mesenchymal stem cell-derived exosomes from different sources selectively promote neuritic outgrowth,” Neuroscience, vol. 320, pp. 129–139, Apr. 2016.

[3]        D. Drago et al., “The stem cell secretome and its role in brain repair,” Biochimie, vol. 95, no. 12. Biochimie, pp. 2271–2285, Dec-2013.

[4]        B. Mendes-Pinheiro et al., “Bone Marrow Mesenchymal Stem Cells’ Secretome Exerts Neuroprotective Effects in a Parkinson’s Disease Rat Model,” Front. Bioeng. Biotechnol., vol. 7, Nov. 2019.

[5]        C. R. Harrell, C. Fellabaum, N. Jovicic, V. Djonov, N. Arsenijevic, and V. Volarevic, “Molecular Mechanisms Responsible for Therapeutic Potential of Mesenchymal Stem Cell-Derived Secretome,” Cells, vol. 8, no. 5, p. 467, May 2019.

[6]        G. Maguire, “Stem cell therapy without the cells,” Commun. Integr. Biol., vol. 6, no. 6, 2013.

[7]        I. Vishnubhatla, R. Corteling, L. Stevanato, C. Hicks, and J. Sinden, “The Development of Stem Cell-Derived Exosomes as a Cell-Free Regenerative Medicine,” J. Circ. Biomarkers, vol. 3, p. 2, Jan. 2014.

[8]        Y. S. Chen, E. Y. Lin, T. W. Chiou, and H. J. Harn, “Exosomes in clinical trial and their production in compliance with good manufacturing practice,” Tzu Chi Medical Journal, vol. 32, no. 2. Wolters Kluwer Medknow Publications, pp. 113–120, 01-Apr-2020.

Reviewer 2 Report

The manuscript clearly summarizes current research on the treatment of neuroregenerative traumatic diseases using MSC exosomes.
It would be useful to discuss more about the route of administration of exosomes in terms of therapeutic efficacy.
The mechanism of action is described in general terms and is not discussed in connection with the composition of exosomes.
The role of protein cargo and overall exosome content in the treatment of neurodegenerative diseases is not discussed.

Author Response

The manuscript clearly summarizes current research on the treatment of neuroregenerative traumatic diseases using MSC exosomes.

It would be useful to discuss more about the route of administration of exosomes in terms of therapeutic efficacy.

Answer:

This is a very important comment.

Although the most popular route of administration is intravenous (IV) injection, it was demonstrated, using an in vivo neuroimaging, that MSC-exo can cross the BBB after intranasal administration more efficiently compares to intravenous injection [1]. This was further supported in near infrared (NIR) imaging also showed that the intranasal administration delivered DiR-labeled MSC-exosomes into the brain, whereas tail vein injection primarily resulted in liver and kidney [2]. Nevertheless, no comparative study was performed to analyze all route of administration in the same model in order to examine functional efficiency.

The mechanism of action is described in general terms and is not discussed in connection with the composition of exosomes.

The role of protein cargo and overall exosome content in the treatment of neurodegenerative diseases is not discussed. 

Answer

Thank you for your comment.

The exact molecular mechanism of action through which MSC-derived exosomes operate is not fully understood due to the wide variety of molecules comprising the exosomal cargo. ExoCarta, an exosome database contains 41,860 protein, 1116 lipid molecule, 3408 mRNA, 2838 miRNA entries, derived from studies of exosomes in several species [3]. Thereby, a variety of functions and multiple molecules can be excreted from exosomal cargo [4]. Among other functions, exosomal proteins can act as signaling molecules, receptors, cell adhesion molecules. For example, the expression of proteins such as nestin, neuro-D, growth-associated protein 43, synaptophysins, VEGF, FGF promote events such as neural development, synaptogenesis and angiogenesis [5]. Moreover, it was indicated that MSC-derived exosomes from adipose tissue contain neprilysin, an enzyme capable of degrading Aβ, and in co-culture with cells designed for Aβ exacerbated production, these exosomes significantly reduced levels of Aβ1–40 and 1–42 [6]. Furthermore, MSC-Exo contain several immunomodulatory factors including transforming growth factor-β (TGF-β), hepatic growth factor (HGF), indolamine 2,3-dioxygenase-1 (IDO-1), interleukin (IL)-10, IL-1 receptor antagonist (IL-1Ra) and prostaglandin E2 (PGE2) [7].

miRNA can also control functions related to neural remodeling as well as angiogenic and neurogenic processes [4]. It has been reported that exosomes also contain miR-98, miR-155 and miR-125a which have antiapoptotic activity [8], [9]. miR-143 and miR-21 , which play an important role in immune response modulation and in neuronal death associated with an environment of chronic inflammation were also found to be present in MSCs-derived exosomes [10]. Similarly, a miRNA cluster formed by miR-17, miR-18a, miR-19a/b, miR-20a, and miR-90a, was also found to be present in MSCs-derived exosomes, and described as important modulators of neurite remodeling and neurogenesis, as well as stimulators of axonal growth and CNS recovery [11]. Overall, there are multiple potential pathways through which MSC-derived exosomes may operate. Nevertheless, as mentioned above, there are several suggested pathways that has been implicated in the exosomal mechanism of action.

This section was inserted within the text (highlighted).

[1]        N. Perets, S. Hertz, M. London, and D. Offen, “Intranasal administration of exosomes derived from mesenchymal stem cells ameliorates autistic-like behaviors of BTBR mice,” Mol. Autism, vol. 9, no. 1, Nov. 2018.

[2]      Y. Liang et al., “Mesenchymal Stem Cell − Derived Exosomes for Treatment of Autism Spectrum Disorder,” ACS Appl. Bio Mater., Aug. 2020.

[3]      H. S. Joo, J. H. Suh, H. J. Lee, E. S. Bang, and J. M. Lee, “Current knowledge and future perspectives on mesenchymal stem cell-derived exosomes as a new therapeutic agent,” International Journal of Molecular Sciences, vol. 21, no. 3. MDPI AG, 01-Feb-2020.

[4]      E. E. Reza-Zaldivar, M. A. Hernández-Sapiéns, B. Minjarez, Y. K. Gutiérrez-Mercado, A. L. Márquez-Aguirre, and A. A. Canales-Aguirre, “Potential Effects of MSC-Derived Exosomes in Neuroplasticity in Alzheimer’s Disease,” Frontiers in Cellular Neuroscience, vol. 12. Frontiers Media S.A., 24-Sep-2018.

[5]      M. Chopp and Y. Li, “Treatment of neural injury with marrow stromal cells,” Lancet Neurology, vol. 1, no. 2. Lancet Publishing Group, pp. 92–100, 01-Feb-2002.

[6]      T. Katsuda et al., “Human adipose tissue-derived mesenchymal stem cells secrete functional neprilysin-bound exosomes,” Sci. Rep., vol. 3, no. 1, pp. 1–11, Feb. 2013.

[7]      C. Harrell, C. Fellabaum, N. Jovicic, V. Djonov, N. Arsenijevic, and V. Volarevic, “Molecular Mechanisms Responsible for Therapeutic Potential of Mesenchymal Stem Cell-Derived Secretome,” Cells, vol. 8, no. 5, p. 467, May 2019.

[8]      X. Cheng et al., “Mesenchymal stem cells deliver exogenous miR-21 via exosomes to inhibit nucleus pulposus cell apoptosis and reduce intervertebral disc degeneration,” J. Cell. Mol. Med., vol. 22, no. 1, pp. 261–276, Jan. 2018.

[9]      J. F. Ma, L. N. Zang, Y. M. Xi, W. J. Yang, and D. Zou, “MiR-125a Rs12976445 Polymorphism is Associated with the Apoptosis Status of Nucleus Pulposus Cells and the Risk of Intervertebral Disc Degeneration,” Cell. Physiol. Biochem., vol. 38, no. 1, pp. 295–305, Jan. 2016.

[10]      S. R. Baglio et al., “Human bone marrow- and adipose-mesenchymal stem cells secrete exosomes enriched in distinctive miRNA and tRNA species,” Stem Cell Res. Ther., vol. 6, no. 1, Dec. 2015.

[11]      H. Vilaça-Faria, A. J. Salgado, and F. G. Teixeira, “Mesenchymal Stem Cells-derived Exosomes: A New Possible Therapeutic Strategy for Parkinson’s Disease?,” Cells, vol. 8, no. 2, p. 118, Feb. 2019.

Reviewer 3 Report

The review by Guy & Offen regarding MSC-derived exosomes is a relatively well written work. The review highlights the main discoveries and the furrent state of the field. The review contains only minor grammatical mistakes and reads well. However, it would benefit from addressing the following points:

1) The abstract and introduction imply that MSC therapeutic effects are modulated primarily by exosomes. There does not seem to be enough evidence in the literature to make this claim.

2) Section 2 states that exosomes play an important role in intercellular communication. Is there enough evidence in the literature to make this claim?

Author Response

The review by Guy & Offen regarding MSC-derived exosomes is a relatively well written work. The review highlights the main discoveries and the furrent state of the field. The review contains only minor grammatical mistakes and reads well. However, it would benefit from addressing the following points:

1) The abstract and introduction imply that MSC therapeutic effects are modulated primarily by exosomes. There does not seem to be enough evidence in the literature to make this claim.

The comment was accepted, and corrected in the text.

2) Section 2 states that exosomes play an important role in intercellular communication. Is there enough evidence in the literature to make this claim?

Thank you for your comment.

Intercellular communication is a term relates to any information exchange between cells.  Although exosomes were originally thought to play a main role in cellular debris disposal, their role in long distance cell–cell communication is nowadays increasingly acknowledged as they function as important transporters of miRNAs, piRNAs, lncRNAs, rRNAs, snRNAs, snoRNAs, tRNAs, mRNAs, DNA fragments, and proteins [20]. This idea was initially demonstrated for Epstein-Barr virus-infected cells, where secreted exosomes transferred viral miRNAs into neighboring non-infected cells, leading to repression of virus-target genes [1]. This was further supported by Meckes et al., in a study revealing that, through exosomal secretion and uptake, a human tumor virus can induce the transfer of a viral oncoprotein, signal transduction molecules, and virus-encoded miRNAs into multiple cell types and activate cell-signaling pathways [2]. Furthermore, after transfer of mouse exosomal RNA to human mast cells, new mouse proteins were found in the recipient cells, indicating that transferred exosomal mRNA can be translated after entering another cell [3]. Following this path, several groups have recently reported that EV-mediated secretion of a given miRNA in the tumor microenvironment is responsible for tumor metastasis [4]. Tumor exosomes were shown to act as a multicomponent delivery vehicle for mRNA, miRNA and proteins to communicate genetic information as well as signalling proteins to cells in their environs [5]. Epithelial exosomes also appeared to be efficient intercellular communication vehicles, allowing protein fragments to escape degradation during transepithelial transport and to transmit immune information to local intestinal immune cells [6]. Moreover, exosomes from Alzheimer patients’ brains contain increased levels of amyloid-beta oligomers were shown to act as vehicles for the neuron-to-neuron transfer of such toxic species in recipient neurons in culture. Blocking the formation, secretion or uptake of exosomes was found to reduce both the spread of oligomers and the related toxicity [7]. In the same manner, exosome-associated α-syn oligomers were shown to be taken up by recipient cells and induce more toxicity compared to free syn oligomers [8]. Overall, it was demonstrated that exosomes contain both protein, mRNA and miRNA, which can be delivered to another cell, and can be functional in this new location.

[1]      D. M. Pegtel et al., “Functional delivery of viral miRNAs via exosomes,” Proc. Natl. Acad. Sci. U. S. A., vol. 107, no. 14, pp. 6328–6333, Apr. 2010.

[2]      D. G. Meckes, K. H. Y. Shair, A. R. Marquitz, C. P. Kung, R. H. Edwards, and N. Raab-Traub, “Human tumor virus utilizes exosomes for intercellular communication,” Proc. Natl. Acad. Sci. U. S. A., vol. 107, no. 47, pp. 20370–20375, Nov. 2010.

[3]      H. Valadi, K. Ekström, A. Bossios, M. Sjöstrand, J. J. Lee, and J. O. Lötvall, “Exosome-mediated transfer of mRNAs and microRNAs is a novel mechanism of genetic exchange between cells,” Nat. Cell Biol., vol. 9, no. 6, pp. 654–659, Jun. 2007.

[4]      M. Tkach and C. Théry, “Communication by Extracellular Vesicles: Where We Are and Where We Need to Go,” Cell, vol. 164, no. 6. Cell Press, pp. 1226–1232, 10-Mar-2016.

[5]      J. Skog et al., “Glioblastoma microvesicles transport RNA and proteins that promote tumour growth and provide diagnostic biomarkers,” Nat. Cell Biol., vol. 10, no. 12, pp. 1470–1476, 2008.

[6]      J. Mallegol et al., “T84-Intestinal Epithelial Exosomes Bear MHC Class II/Peptide Complexes Potentiating Antigen Presentation by Dendritic Cells,” Gastroenterology, vol. 132, no. 5, pp. 1866–1876, 2007.

[7]      M. Sardar Sinha et al., “Alzheimer’s disease pathology propagation by exosomes containing toxic amyloid-beta oligomers,” Acta Neuropathol., vol. 136, no. 1, pp. 41–56, Jul. 2018.

[8]       K. M. Danzer et al., “Exosomal cell-to-cell transmission of alpha synuclein oligomers,” Mol. Neurodegener., vol. 7, no. 1, p. 42, Aug. 2012.

Round 2

Reviewer 1 Report

All the comments were addressed by the authors.